# Digital Strategies to Enhance Cultural Heritage Routes: From Integrated Survey to Digital Twins of Different European Architectural Scenarios

Sandro Parrinello [1] and Francesca Picchio [2,*]

1   Department of Architecture, University of Florence, 50121 Florence, Italy; sandro.parrinello@unifi.it
2   Department of Civil Engineering and Architecture, University of Pavia, 27100 Pavia, Italy
*   Correspondence: francesca.picchio@unipv.it

**Abstract:** This paper focuses on a research project for the acquisition and post-production of digital data to create informative virtual representations and digital twins of different European Cultural Heritage sites. The goal was to establish a reliable database for a multi-scalar web platform, also accessible through extended reality (XR) tools. This initiative aims to support the promotion and management of cultural and historical monuments within the context of European Cultural Routes supported by the Council of Europe. The project involves different case studies spanning European geographic regions, such as the Upper Kama in Russia, the Valencian Routes of Jaime I in Spain, and the Gdańsk fortresses in Poland. The methodology employed in this effort primarily relies on integrated rapid survey techniques. Unmanned aerial vehicles (UAVs) and simultaneous localization and mapping (SLAM) technologies were used for data collection. These methods contribute to the creation of accurate 3D databases and models that transform the cultural routes into a digital format accessible via an informative platform. The actions presented in this paper are part of the European project "PROMETHEUS", which is funded by the Horizon 2020 program of the European Union. The project involves collaboration between universities and enterprises, fostering inter-sectoral cooperation. Various techniques such as photographic archives, census analysis, and scan-to-BIM (building information modeling) processes are employed to develop this method further. In fact, the ultimate goal of the project is to establish a framework that can be replicated in other cultural contexts, enhancing the digital documentation and valorization of heritage sites.

**Keywords:** integrated survey; digital documentation; UAVs; scan-to-BIM; digital twins; extended reality; cultural routes; interactive informative platform; H2020 Prometheus



## 1. Introduction

The recognition and protection of cultural heritage routes in Europe have stimulated the development of new approaches and guidelines focused on preserving the intricate relationships between the sites involved [1,2]. This entails a deeper understanding of the tangible and intangible connections that exist within a specific territory. By comprehending the historical, social, and cultural context of these routes, it becomes possible to safeguard their unique characteristics and cultural identity [3].

Therefore, it is essential to document and preserve the historical and architectural values associated with each element of the route. By highlighting these values, it becomes possible to raise cultural and tourist awareness about the entire route on a regional scale, enhancing its significance and appeal to visitors. Furthermore, as Cultural Heritage routes continue to evolve, it is essential to adopt dynamic intervention protocols that move away from static approaches and embrace more flexible and adaptive methodologies. This transformation in investigation methods includes the documentation of heritage, the development of dynamic digital archives, and the establishment of standardized languages to communicate the material and immaterial values of the route.

Qualifying a cultural heritage route corresponds in a way to defining a landscape. It is a matter of setting limits to something indefinite, to include textiles of a mosaic that at first does not appear clear, but whose beauty can only be perceived. Each portion of this mosaic must be framed, analyzed, and defined, and then be positioned so as to compose a new design of the original landscape phenomena. It certainly does not express the same image as the real one, but thanks to new technologies and digital applications, certain aspects of its complexity can be enhanced. Cataloging the acquired information and interconnecting metadata and databases enable the creation of complex systems of models and digital products that elucidate the elements of the route. To manage this multitude of products, systems that support the interpretation and understanding of the technical and historical–cultural features become necessary. By analyzing these features, intervention and implementation procedures can be devised to establish connections between the past, the present, and the future of a cultural route or its associated architectural elements.

This holistic approach involves comprehensive documentation, the preservation of cultural identity, and dynamic intervention protocols to describe each route. Integrated surveying technologies and methodologies, coupled with digital databases and standardized languages, play a vital role in achieving these objectives. They are employed to create digital databases and digital twins (DTs) that encompass detailed and extensive representations of the heritage [4]. By leveraging these data, DTs can provide virtual replicas of architectural environments, complete with historical and cultural information, technological and constructive aspects, material and immaterial values are all datasets that are useful to connect a virtual model (digital) with its physical entity (real) [5].

The research presented in this paper addresses the issue of insufficient documentation and subsequent undervaluation of the inherent cultural values associated with historical routes. This deficiency leads to a gradual detachment from the tangible heritage, underscoring the necessity for a fresh approach to represent these sites. This approach should serve as both a vessel and a source of information encompassing architectural assets and the surrounding landscape. In this sense, digital twins serve as a solution, representing highly accurate and morphometrically precise 3D representations of these monuments. By enriching these models with specific content, they establish a connection with the heritage they portray, unveiling aspects that might not always be readily apparent, e.g., past configurations or possible future developments on the artefact.

The data gathered and employed in constructing a DT of an architectural artifact encompasses not only its architectural intricacies, but also its historical and cultural significance within the wider context of its surroundings. This extends to its interplay with nearby monuments, regardless of their inclusion within the designated route. Consequently, the amassed information not only ensures a parallel and verifiable alignment with the physical object being replicated through the digital twin, but also establishes correlations with analogous artifacts that share proximity in both geographic and thematic terms (date of build, functions, constructive materials, style, etc.).

The strategy employed to generate these digital twins embraces an all-encompassing and multidisciplinary documentation methodology. The resilience of this experimental methodology is rooted in its pragmatic procedural frameworks, which have been derived from a series of field tests conducted over diverse pilot cases. This lends the methodology a heightened efficacy and a capacity for seamless adaptation and repeatability to diverse contexts for future implementation.

First of all, adopting expeditious survey methodologies and integrating heterogeneous data was used to create reliable databases for 3D models for all analyzed contexts. Consequently, different modeling strategies (parametric modeling, reverse-modeling, and mathematical–geometric modeling) were adopted according to the different descriptive objectives of each monument and each of the analyzed routes. Lastly, the project culminated in the establishment of a digital user–space interaction system, achieved by developing 3D interpretative web-platforms dedicated to cultural routes.

The experimentation of this methodology inside PROMETHEUS H2020 has been conducted by 35 international researchers, affiliated with different universities (University of Pavia, University of Florence, Polytechnic University of Valencia, Gdańsk University of Technology, and Perm National Research Polytechnic University) and several enterprises (EBIME, CTA, SISMA, BLESARQ, and Metaheritage) (Figure 1). The objective of this extensive collaboration was to formulate collective strategies for constructing comprehensive architectural digital 3D models. These models aimed to encapsulate the architectural heritage's evolution across its specific route, encompassing its present, historical, and potential future configurations.

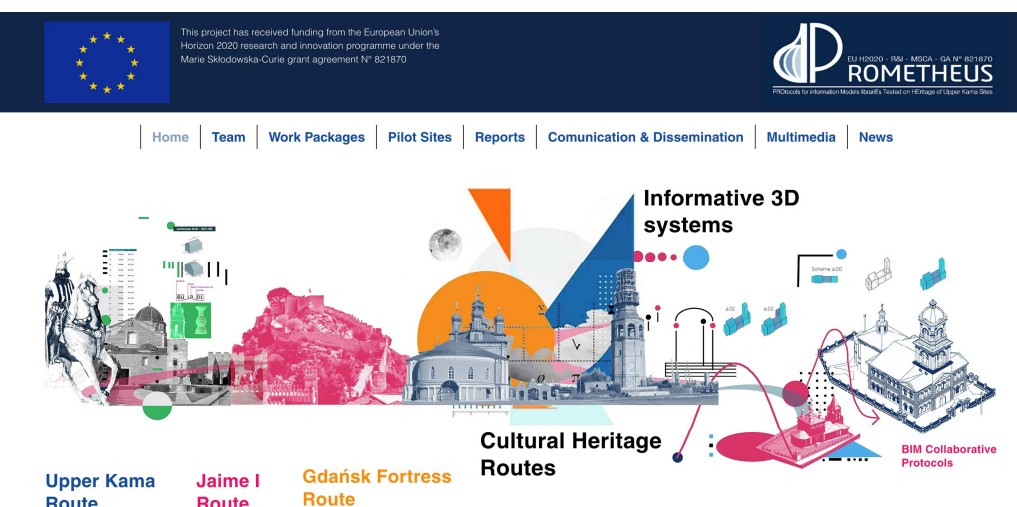

**Figure 1.** H2020 PROMETHEUS website home page (Graphic Elaboration by A. Dell'Amico).

Pilot cases have been selected always considering their scale and extension, aiming to develop specific knowledge strategies for each route and define useful designs for the digital storytelling of this tangible and intangible heritage. The results generated, comprehensive 3D multi-scaled models, connected information, and virtual platforms in which this connection happens, enable immersive and interactive experiences, allowing visitors to explore and engage with the heritage sites virtually. New ways of interactive communication in the domain of multimedia Cultural Heritage applications and the development of advanced technologies are facilitating access and increasing the value and public awareness of Cultural Heritage [6]. Augmented reality and virtual reality can be integrated to offer enhanced storytelling and educational experiences [7], bringing the past or future of currently unknown and endangered routes to life in a dynamic and accessible way, also ensuring their longevity and accessibility for future generations [8].

## 2. Materials and Methods

### 2.1. Site Description

Analyzing cultural landscape contexts differing in history, culture and architectural typology becomes fundamental to structuring a methodological protocol for the acquisition and processing of heritage information platforms. This analysis helps in understanding the different requirements and challenges associated with acquiring and processing heritage information.

The application was undertaken on some European contexts—identified in the project partner universities—which are heterogeneous in terms of the type of Cultural Heritage routes and investigated at different scales. These contexts included the widespread heritage of Upper Kama (Russia) at territorial scale, the Jaime I sites (Spain) at the provincial scale, and the fortification system of the city of Gdańsk (Poland) at the urban scale [9] (Figure 2).

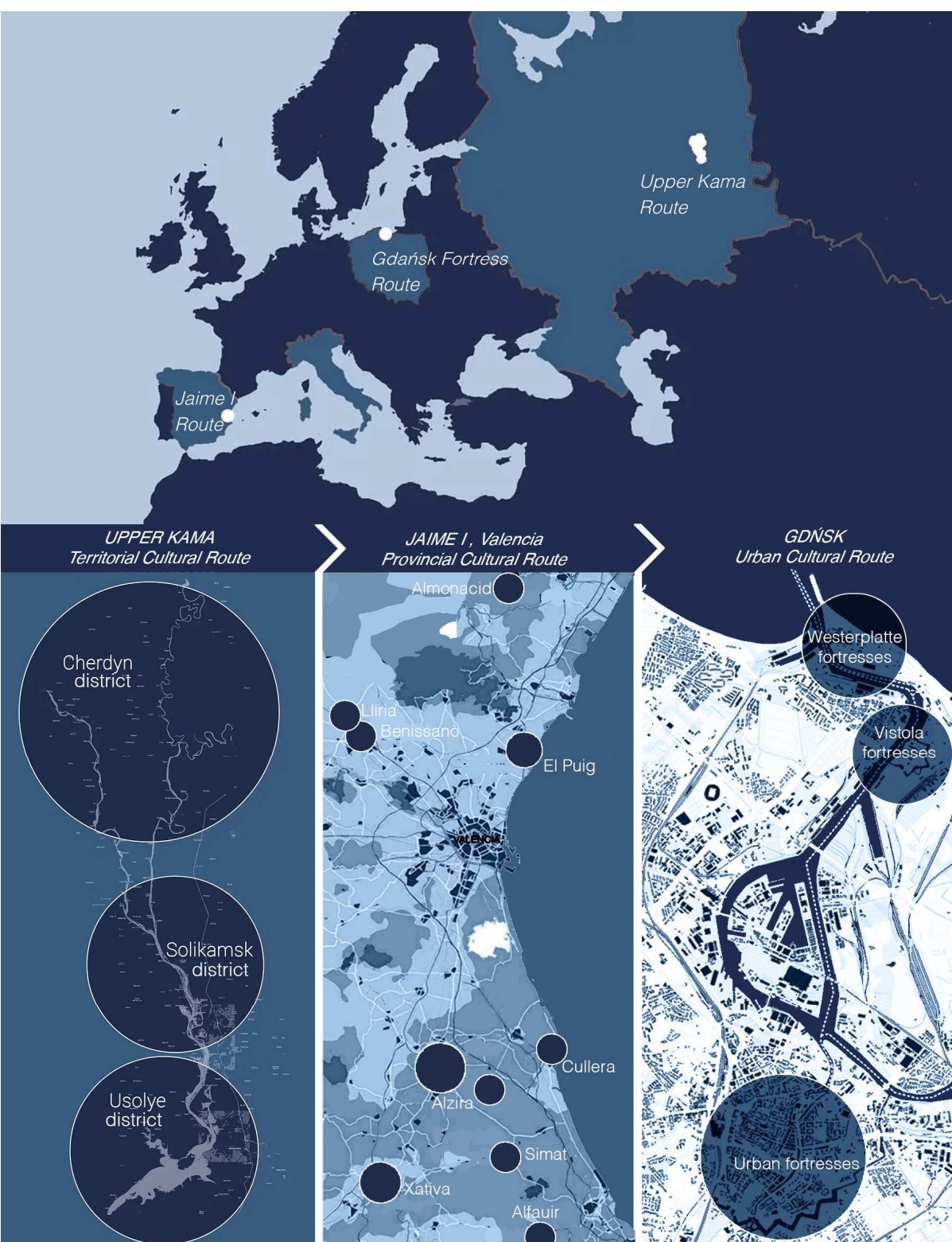

**Figure 2.** European Cultural Routes: territorial level (Upper Kama), provincial level (Valencia), and urban level (Gdańsk).

These are three case studies with different extensions, and which bring together complex architectural phenomena in which the value of the cultural route is fully manifested. First and foremost are the cultural–historical influences. In these case studies, the tradition, the antecedent—be it the Tartar, Moorish, or Teutonic Knights' pre-existences—is modified

as a result of a series of historical events that give rise to cultural contaminations. These contaminations modify language, decoration, and stylistic models, producing new forms and patterns or, more simply, promoting the development of signs that renew and characterize the landscape. Signs becomes historicized and becomes an unforgettable part of a cultural identity, definitively modifying the place and conditioning the events that mark, from that moment on, the future of that particular socio-political and economic context. In addition to influences, secondly, time must be considered, both the time in which these architectural evidences are built, and the time of propagation of the cultural phenomenon, the stability or invariance of the same influences in the different time frames. Time becomes an essential reference factor for the classification of the route, because it allows us to determine limits and order the events that have characterized the history of these landscapes.

- The Upper Kama region is located north of Perm Krai, limited to the west by the chain of Ural Mountains, and to the north by the Komi region. Since the 15 century, the salt trade, from Europe beyond the Urals to China [10], has increased trade flows and intermediate settlements along this route, centered around the main districts of Solikamsk (1430), Cherdyn (1535), and Usolye (1606) [11,12]. The Upper Kama region is still composed of more than 50 monumental Orthodox complexes, spread over the territory but interconnected by a single cultural language, but where management planning aimed at protecting these assets is absent.
- The route of Jaime I in Spain is a cultural itinerary that runs through those territories affected by the Catholic Reconquista of 1238 and represented by the current Provinces of Valencia, Castellón, and Alicante [13]. Although there is no real official source on the itinerary associated with Aragonese King Jaime I, the tourist and cultural promotion of this route is aimed at ideally connecting the 112 sites (religious buildings, defensive and strategic buildings, and some minor buildings) located between Teruel, where the itinerary begins, and Peñíscola, where it ends. The selection rationale of the sites investigated was dictated, on the one hand, by their consistency from a historical, geographical, and architectural point of view with the various strands belonging to the route; on the other hand, by the numerous transformations many of them have undergone over time, which have led to their progressive deterioration, abandonment and, in some cases, consequent ruin.
- The fortifications in Gdańsk define an organic system, closely integrated with the city, but defining a clearly visible spatial boundary. This system consists of fortifications dating from different historical periods and therefore classifiable typologically and temporally into macro-groups (from the 10th century to the 20th century): line fortifications from the medieval period, early modern outer fortifications, and 20th century fortifications, erected and expanded to defend the New Port and the Westerplatte peninsula [14–16].

Documentation of the image of a place and its monuments, framed in their current condition, becomes an objective of territorial monitoring of the evolutionary processes that involve culture and local identities [17], and it defines a path of initiative and evolution for the development of adequate maintenance and management plans [18].

The primary objective of the project was to represent the complex relationships that define the identity of these places, including the monuments and their connections to different cultural contexts, and thus, the 'phenomenon' of the specific cultural route, through a specific and new digital configuration. The digitization process served as a methodological approach accompanied by a thorough analysis involving the decomposition and critical reconstruction of the architectural units. This was achieved through the semantic classification of its sub-elements—from landscapes to buildings, and from the buildings to their details—in all their relationships with the contexts [19] (Figure 3).

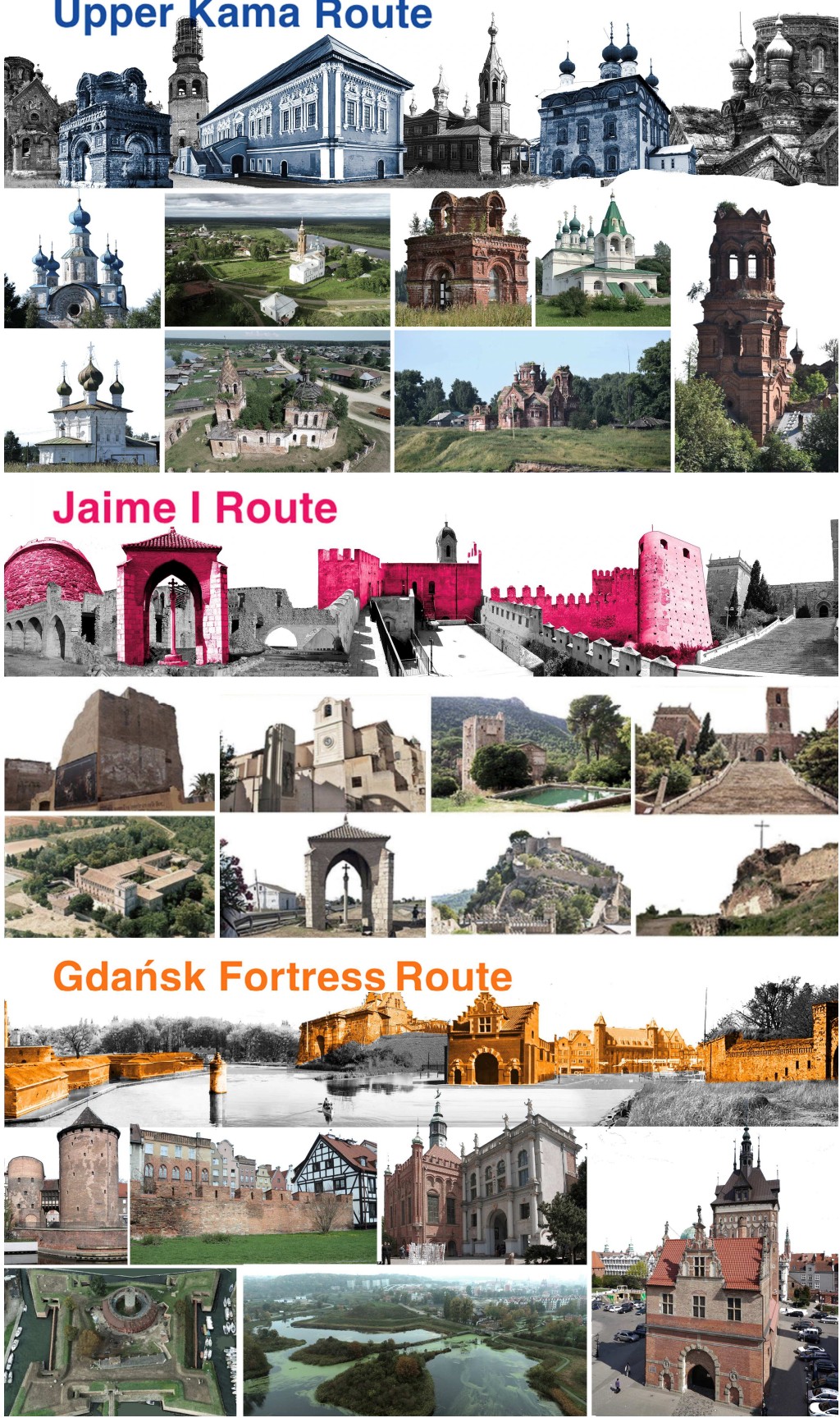

**Figure 3.** Images of some of the investigated sites along the routes.

In the case of the Upper Kama, a large part of the investigation involved verifying the existence of monuments and finding them, as sometimes this built heritage is hidden in the forests, within an extremely large area. In the other two cases, the work was very different. For the Spanish route, the interaction with colleagues from the Polytechnic University of Valencia and the presence of numerous studies on the route facilitated the recognition of the sites and the organization of the maps that identified the monuments to be considered.

For the Gdańsk fortresses, on the contrary, there was no unified study that intended to analyze the defensive heritage according to cohesion logic with respect to the evolutionary grid of the city. Interaction with archives and colleagues in the history of architecture at the Polytechnic University of Gdańsk was therefore crucial in order to be able to systematize each monument, and then to standardize the language of communication with that of other case studies [20].

### 2.2. Integrated Survey and 3D Databases

Acquisition and survey campaigns were conducted to create digital duplicates of the monuments along the different European context routes. These duplicates served as the base for a detailed analysis of the historical and evolutionary phases of each church, castle, or fortress.

Several tools were employed to produce a digital reference archive of the city and its historical–architectural image. Aiming to define repeatable acquisition protocols for the development of 3D models of the built heritage, all the case studies included the testing of expeditious documentation methodologies. These actions included the joint application of image-based and range-based instruments whose data, suitably integrated, contributed to the production of three-dimensional multi-scalar databases [21], from which it was possible to understand and interpret the digitized cultural route: from the building to the territory and from the territory to the building. An integrated survey methodology was adopted, utilizing terrestrial laser scanners (TLS), mobile laser scanners (MLS), and terrestrial and aerial photogrammetry (UAV). This strategy ensured the acquisition of accurate spatial data and photographic information, which was crucial for structuring and validating the geometric database [22] (Figure 4).

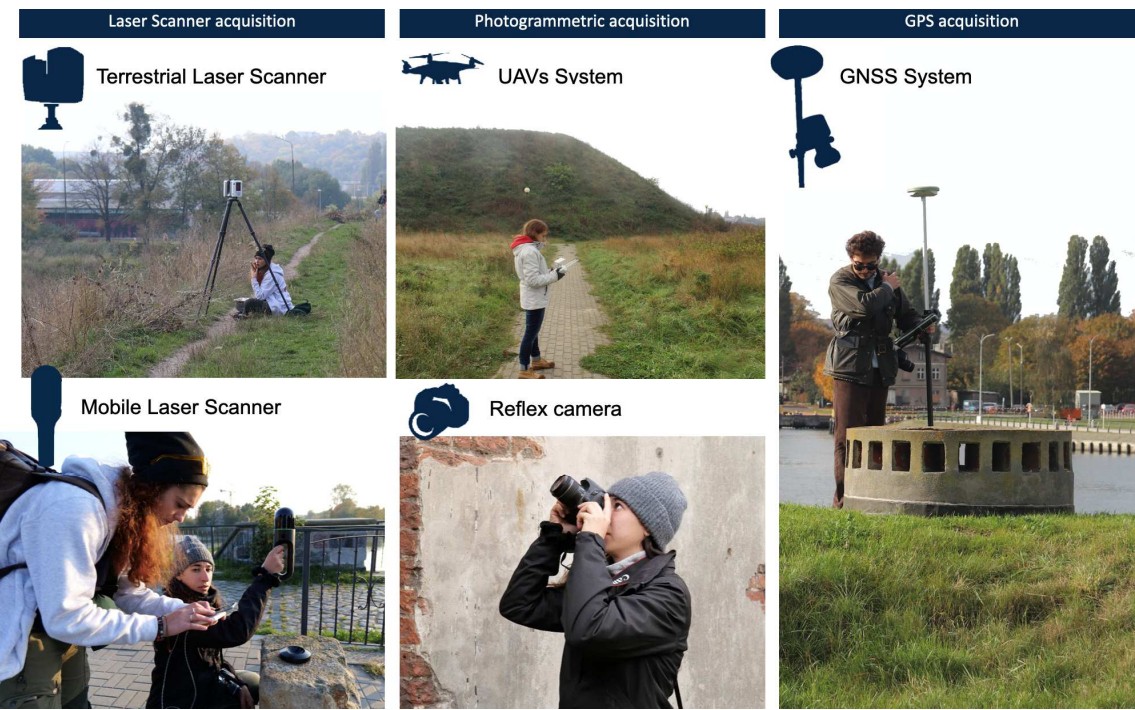

**Figure 4.** Different digital tools used for data acquisition.

Furthermore, the use of rapid-survey instrumentation was necessary to understand the extent of the three routes and to ensure the replicability of the method [23], both in terms of timing and the type of product obtained. The archives obtained with these tools and other data (3D panoramas, high-resolution landscape images, etc.) provided the necessary information to ensure the reliability, in terms of geometric and colorimetric accuracy, of the architectural models.

The first experience gained from the Upper Kama territory provided valuable insights and knowledge that guided the acquisition standards and methods for the other European contexts. The documentation methodology was adapted to specific typological features, enabling efficient and expedient data acquisition. The photogrammetric tool, both from close-range and UAVs, was tested to generate comprehensive 3D models capable of effectively describing each monument [24,25] (Figure 5).

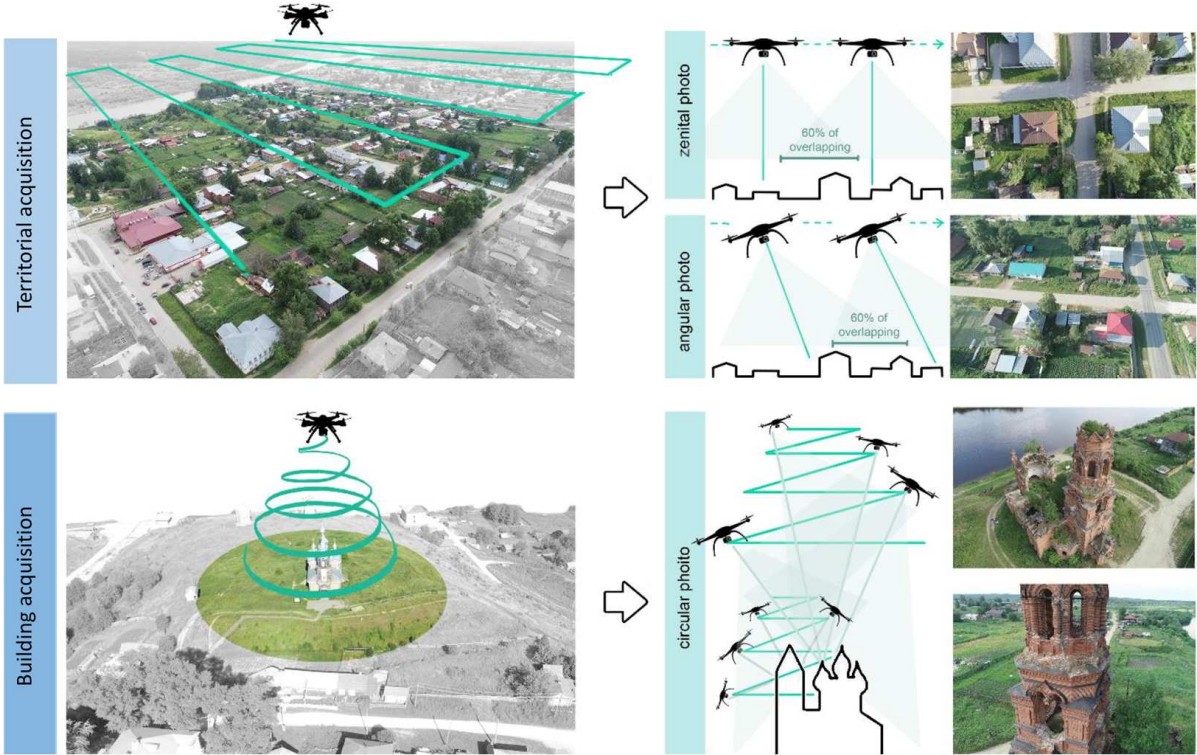

**Figure 5.** Scheme for close-range data acquisition with UAVs designed for the Upper Kama context.

The use of drones was crucial to acquire data from inaccessible areas or architectural surfaces (high heights of some vertical elements, interdicted areas that are not visible from the driveways or pedestrian streets roads, the presence of construction sites or debris in the proximity of some monuments, etc.) and ensure uniformity in describing each analyzed complex. The georeferenced database of the route obtained by drone in some cases enabled the georeferencing of individual architectural databases obtained with LIDAR instruments. For this reason, the drone was used at an altitude 30 to 60 m maximum above the ground that ensured a high level of resolution of the acquired images (designed GSD of 1–2 cm/pixel maximum) (Figure 6).

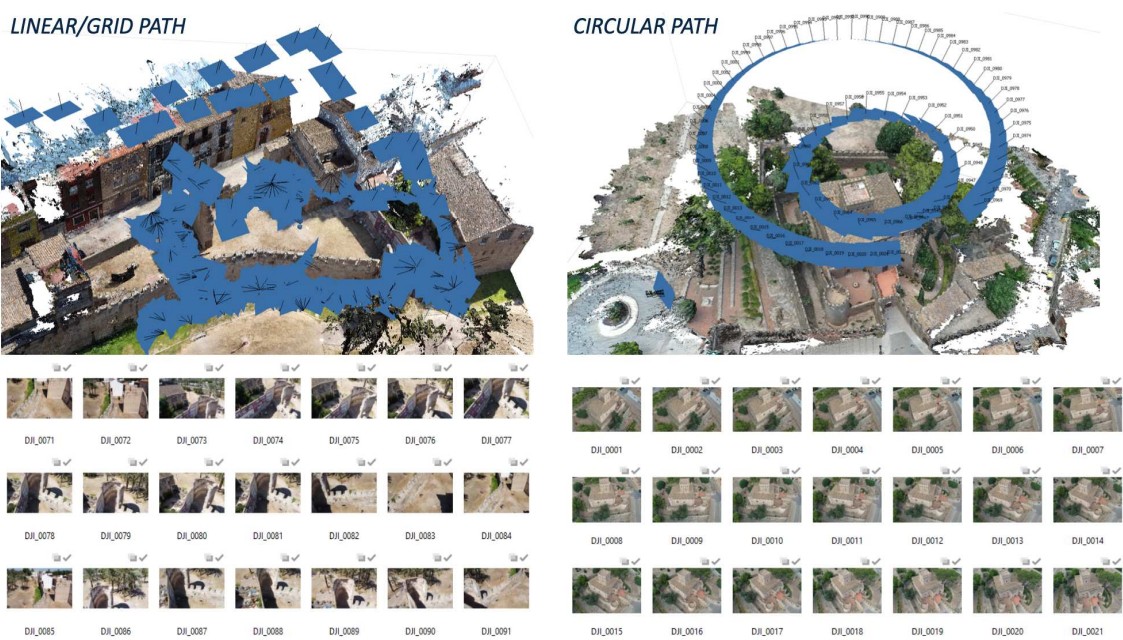

**Figure 6.** UAV acquisition with DJI mavic mini drone in Valencia province.

In cases where certain areas were not accessible by drone due to several factors, such as tree cover or flight restrictions, mobile systems were utilized for the rapid interior surveys of buildings and those areas. Close-range photogrammetry complemented the documentation process (Figure 7), focusing on surfaces requiring a more detailed description, especially for masonry textures or areas with specific decay conditions.

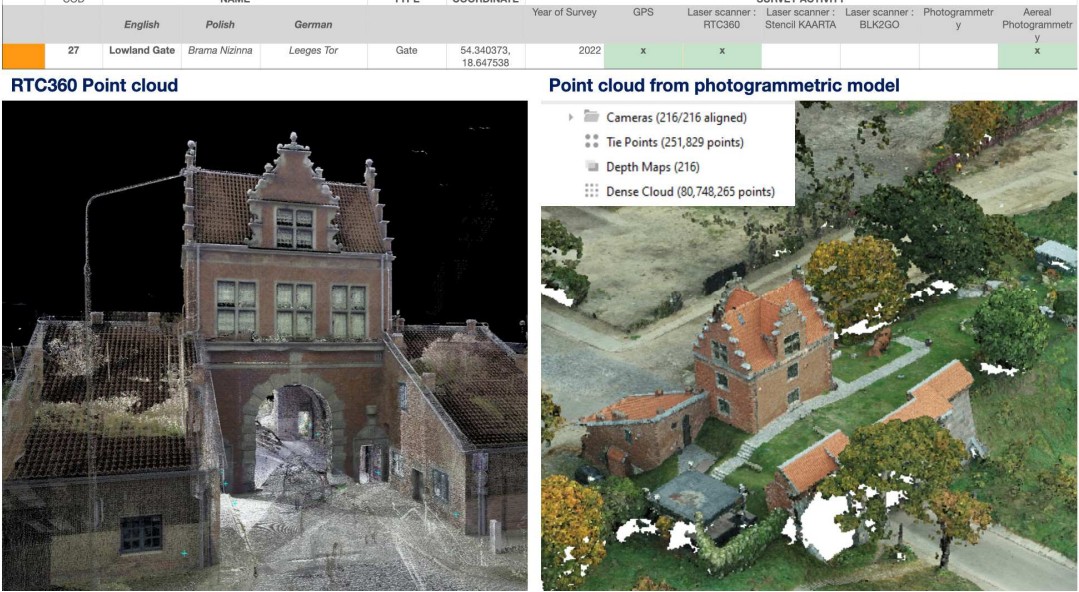

**Figure 7.** Integrated data acquisition of Lowland Gate (Gdańsk). On the left, Laser Scanner RTC360 point cloud; on the right, point cloud from UAV DJI Phantom RTK.

To achieve the project's objectives, it became necessary to assess and compare the effectiveness of various digital survey methodologies. The aim of creating 3D archives and informative models of heritage required the definition of a specific methodology that balanced the principles of metric reliability and survey precision with procedural efficiency.

Being "swift" within the available time on site proved crucial both for the specific project and in attempting to envision replicable procedures in other cultural contexts.

Specifically, the methodology aimed for data integration, creating a comparative matrix among different types of measurements, information, and metadata, describing and constructing a heterogeneous database for each monument at various levels. The diversity of data, in relation to the diversity of analyzed architectures, becomes functional both for evaluating process replicability and for the proper functioning of the interactive platform, the model utilization system.

Digital surveying, despite striving to provide a wealth of information about locations, is always directed toward a specific purpose. The interactive platform, the digital twin of the cultural route, must consider numerous aspects in which the acquired information's specificities take on different connotations—designs, photographs, 3D models, texts, etc. Surveying and modeling design must always compare the real dimensions of the cultural asset, the one that is surveyed, with its digital translation, envisaging a distinct dialogue between these two images.

The cultural landscape expressed by the routes is thus narrated through dynamic databases that contribute to updating the management and valorization systems of the built environment. It is precisely in the interaction with digital data that the specificity of its informational potential must be characterized. For instance, the management aspect is quite distinct from the enhancement and the various ways in which heritage can be valorized, both in terms of increasing awareness and in explicating the historical events that have shaped its evolution.

In this regard, 3D models must be capable of gathering all the necessary information to enable freedom of utilization coupled with understanding.

In order to structure a modeling strategy that could respond to the multiple descriptive needs mentioned above (knowledge for conservation, informatization, and alternative fruition of the route), different scan-to-modeling processes [26,27], also characterized by different levels of development (LODs), were tested at the urban scale for the methodology of data elaboration.

First of all, four levels of interpretation and LODs were defined:

1.  Territorial level: At this level, different centers within the districts are identified using a geomatic hyper-simplification model of the landscape, territory, infrastructure, and connectivity systems, as well as the potential service systems that can be linked to this scale. Specifically, during this phase, the development involved creating a simplified digital terrain model (DTM) with the main contour lines based on existing cartographic data and metric information obtained from integrated surveys (utilizing tools such as mobile laser scanners and UAVs). This model delineated the connectivity and accessibility attributes of individual monuments, outlining potential connecting pathways between religious and fortified complexes. This corresponds to LOD 0 (digital terrain model) with an accuracy level of approximately 10 m [28].

2.  Urban level: In this model, all the different activities and services present within the single center are identified. This level corresponds to LOD 1 (one-block model), with an approximate accuracy of 5 m.

3.  Monumental area level: Within the single urban center, the model of the monumental area will be deepened, in order to identify the different areas for safeguarding the historical character of the single territorial centers [29]. To this end, the landscape features of the area identified in relation to the aesthetic monumental elements within which it will be possible to identify the typological features of the monumental structures present in the area will be described in a single model. This level corresponds to LOD 2 (architectural scaled model), with an approximate accuracy of 3 m.

4.  Monumental building level: The various buildings present in the monumental areas, identified by the previous level of analysis, will be further investigated in this level of investigation where they will be individually described through the construction of a specific HBIM information model in which the decorative and technological

features of the entire building will be represented. This level corresponds to LOD 3 (full architectural scaled model), with an approximate accuracy of 0.2 m.

Subsequently, other types of 3D modeling were developed, including a reverse-modeling approach (surface mesh and surface NURBS), aiming to obtain heterogeneous outputs for experimenting with different cognitive and communicative phases of the route. In reality, although these modeling methodologies were tested on various case studies to create different informative models for the three routes, some of these modeling techniques have been integrated and interchanged, often producing outputs and results that bridge between HBIM (historic building information modeling) and GIS (geographic information systems) informative models, virtual reconstructive models of historical phases, and navigable point clouds, among others.

## 3. Results

### 3.1. Post-Processing Data and Digital Twins

The amount of information acquired through the use of digital survey tools on the three cultural routes was massive. Detailed point clouds on the buildings were flanked by more sparse—but still highly descriptive—point clouds of the landscape in which the monuments were located (Figure 8).

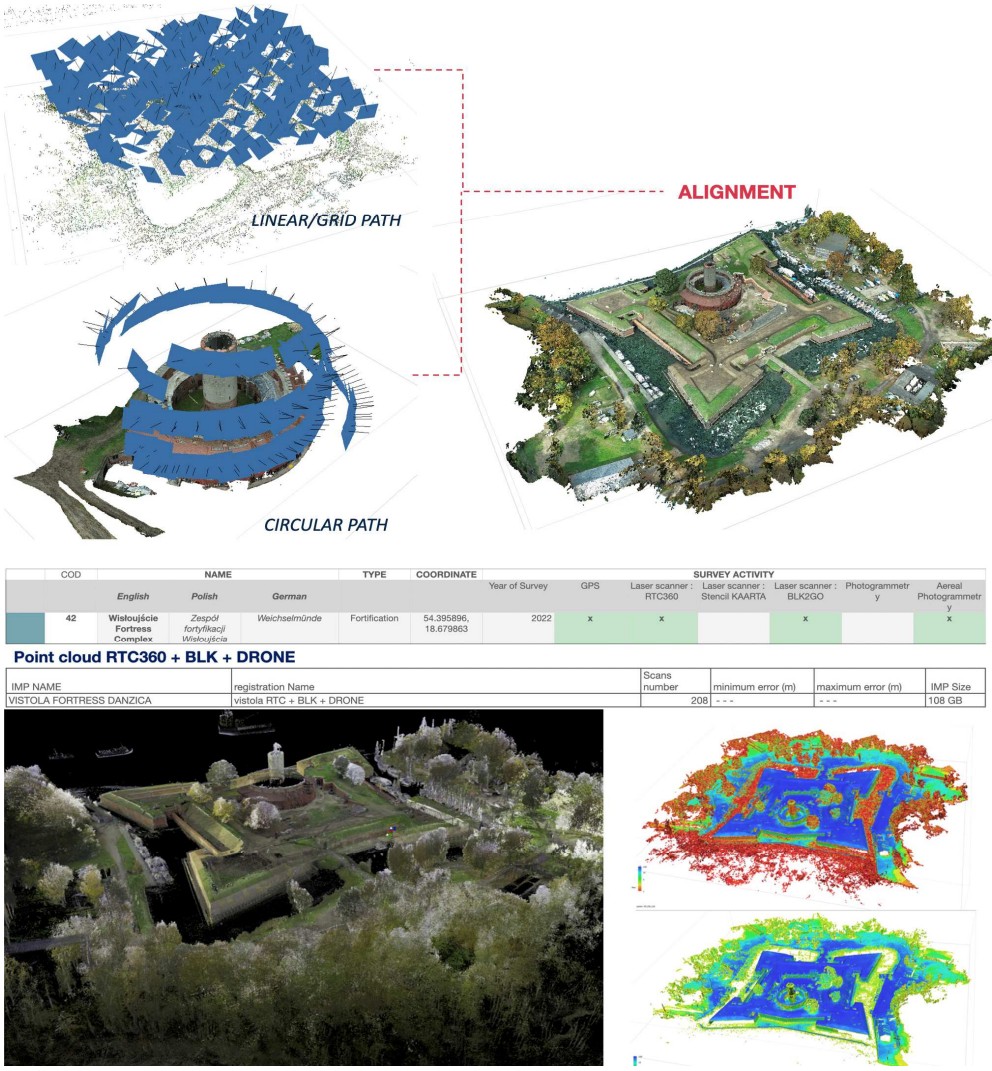

**Figure 8.** Data acquisition of Vistola Fortress (Gdańsk). Above, UAV data integration of linear and circular path acquisition; below, data integration between MLS, TLS, and UAVs.

### 3.1.1. Upper Kama 3D Modeling

Different modeling methodologies were attempted to achieve such levels of detail and test the fruition of the obtained 3D models [30]. The scan-to-BIM process is a commonly used procedure to convert point cloud data into informative 3D models [31]. It involves capturing the geometric and spatial information of a physical structure using 3D laser scanning or similar technologies and then using those data to create a building information model (BIM). The BIM represents, in this sense, a very accurate picture of the geometry, materials, and other relevant information of a building, enabling analysis, visualization, and management of the structure.

Regarding the Upper Kama route, the scan-to-BIM process was extensively tested. The architectural richness of the decorative element of the buildings along this route presented a complex case for experimentation with the simplification of 3D parametric models. The experimentation was tested to reduce the complexity of the models (decimation), while retaining the necessary information by preserving the overall shapes and structures. This simplification helps to optimize the performance and usability of the models, making them more manageable for analysis, visualization, and other applications [32] (Figure 9).

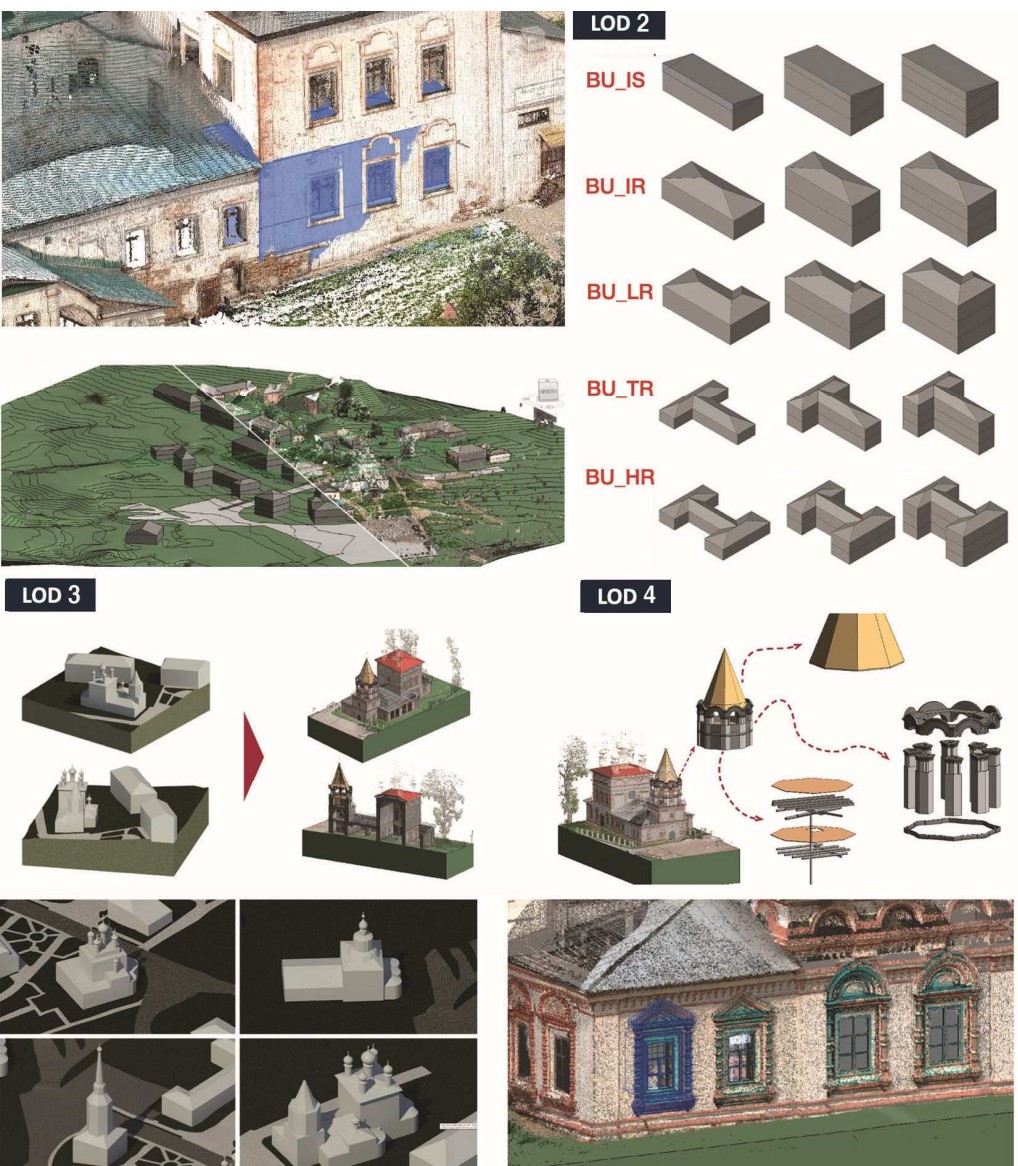

**Figure 9.** Scan-to-BIM process for Upper Kama informative models (data elaboration performed by H. Fu and A. Dell'Amico).

For the description of the spatial, landscape, and urban character levels, the volumes and encumbrances of the buildings were modeled in simple volumes, useful for defining the dimensional features of the encumbrances and orientation [33]. These were then placed on a simplified DTM, modeled from a photogrammetric cloud obtained by drone. The technological and decorative features of some monumental buildings were then explored in a second level of modeling. This branched system allows querying by levels, from the general to the particular, with a logic of the simplified reading and gradual deepening of information related to route knowledge. Each modeled element, both of the landscape and the monument, was imputed into a shared abacus with the aim of building an initial library of digitized elements of the architectural features identifying the monuments of Upper Kama. The modeling phase of the monuments was set in worksharing mode on a scan-to-BIM basis. This process passed from a preliminary phase of recognition and decomposition of the different hierarchical components that constitute the building and that defined the parametric digital twin [34,35].

The information components that qualified the models as digital twins of specific Upper Kama districts routes were a result of the analysis and instrumental acquisition of the survey activities and the filling of the thematic sheets. The sheets were organized according to a structure that included both general information (for recognizing the architectural unit) and morphological–spatial characterization (specific to the volumetric and overall distribution system, facades, architectural details, down to the specifics of ornamentation), as well as historical considerations (linked to archival information), the state of conservation, and the technological characteristics of the building (encompassing both construction aspects related to masonry and the pathologies of present deterioration).

These data engage in a continuous dialogue between what is presented in the model, which explicitly conveys the history that can be discerned from the analysis of architectural elements, and the information provided in the sheets. The digital twin speaks of itself and how the building has come down to the present day, while also envisioning the potential for further development in the future. The digital twin should exist in parallel to the real one, with the capability to transform and update itself as the real structure evolves. In the case of the Upper Kama project, out of the 80 sites surveyed, over 50 have been digitally recorded, with 7 parametrically modeled in BIM following the aforementioned procedures.

### 3.1.2. Jaime I 3D Modeling

Similarly, reverse-modeling procedures were tested from integrated photogrammetric–lidar databases [36]. This procedure was mainly tested for the Jaime I route, where the aim was to produce, in a short timeframe, databases that were exhaustive but at the same time not overloaded with information. The models obtained consisted of geometric and colorimetric information that guaranteed the descriptiveness of the mesh models obtained. These, decimated in the number of polygons to be more easily manageable, nevertheless maintained a high level of realism that can be shown within visualization platforms (Sketchfab) [37]. However, to ensure that these models were equally informative representations of the route, additional information sheets or descriptive elements needed to be incorporated into the systems. These supplementary components can be appropriately linked to the entire model or specific portions of it, enhancing the understanding of the artefact and its connections with the surrounding environment.

For the Jaime I route, over 20 sites have been identified, including castles, city walls, religious buildings, and minor architecture. Integrated and efficient digital surveys were conducted for these, with the aim of subsequent reverse-modeling procedures. Notably, the sites along the Valencian route are larger in comparison to the Russian route or even the Spanish route, which does not exhibit a constellation of isolated episodes across an extensive territory. Instead, it comprises sites of memory that blend into a more intricate landscape system. While most Russian sites consist of scattered episodes, in the Spanish case, the castles, ruins, and fragments of James I's history are found within large architectural complexes that have grown in size and territorial significance over centuries. This is

why the three-dimensional databases and models aimed to more accurately represent the surrounding territory of each individual monument, often embedded within historic centers.

Throughout the stages of data acquisition, development of the digital archive, and representation of the route through various illustrative systems, this specific provincial dimension of Valencia influenced the different methodological choices. This research project was designed to foster dialogue among researchers; therefore, alongside the scientific goals suggested by the proposal, the Spanish experience led to intense debates where two cultural dimensions (Italian and Spanish), historically accustomed to dealing with architectural heritage, sought compromises to construct a language that transcended national ideological boundaries in representation and graphic expression. For this reason, the organization of historical–constructive, technological, and conservation-related information was structured to constitute the informative framework of the realized reverse-modeling models. These models, appropriately semanticized first in their historical–constructive phase, and then in their technological components, are queryable (by volumes or surfaces) and linked to respective infographic information within a simplified interactive platform (a web-GIS system for interaction and Sketchfab for individual model visualization).

By including information sheets or descriptive apparatuses, viewers or users of the models can access more specific knowledge about the route, such as historical context, architectural details, cultural significance, or any other relevant information. These additional resources contribute to a deeper understanding and appreciation of the artefact being represented, enhancing the overall experience and educational value of the models [38].

Apart from visualization, another practical use of these models is the ability to physically reproduce these monuments through solid prototyping. The mesh models derived from point clouds were appropriately optimized, including decimation where necessary, and filling in any portions that may not have been fully captured by digital instrumentation. This optimization process prepares the models for further processing through 3D printing, enabling the creation of physical replicas of the monuments for an alternative and more inclusive fruition of CH routes (Figure 10).

### 3.1.3. Gdańsk Fortresses 3D Modeling

The last modeling strategy from databases is the mathematical tool that uses non-uniform rational basis spline (NURBS); it develops 3D geometric surfaces from 2D vector data obtained by detailed drawings. This process was mainly tested for the last route, that of the Gdańsk fortresses. The decision to operate in this mode in this context was dictated by several factors: on the one hand, for a logistical and temporal question of model development; on the other hand, for methodological experimentation to be carried out on the research output (Figure 11).

As far as the first point is concerned, it should be emphasized that the logistical events connected with the project (the initial blockade for COVID-19 and the subsequent war between Russia and Ukraine) forced us to reshape both the partnership and the subject of the research. In place of the Perm Polytechnic University (Russia) from 2023, less than a year after the project's closure, a new academic partner (the Polytechnic University of Gdańsk) and a local company (CTA) were involved. The timeframe for the development of information models on this route was drastically reduced, forcing a more flexible and less time-consuming methodology to be structured. The idea was to create easily usable models from a navigable information platform (embedded in a GIS environment).

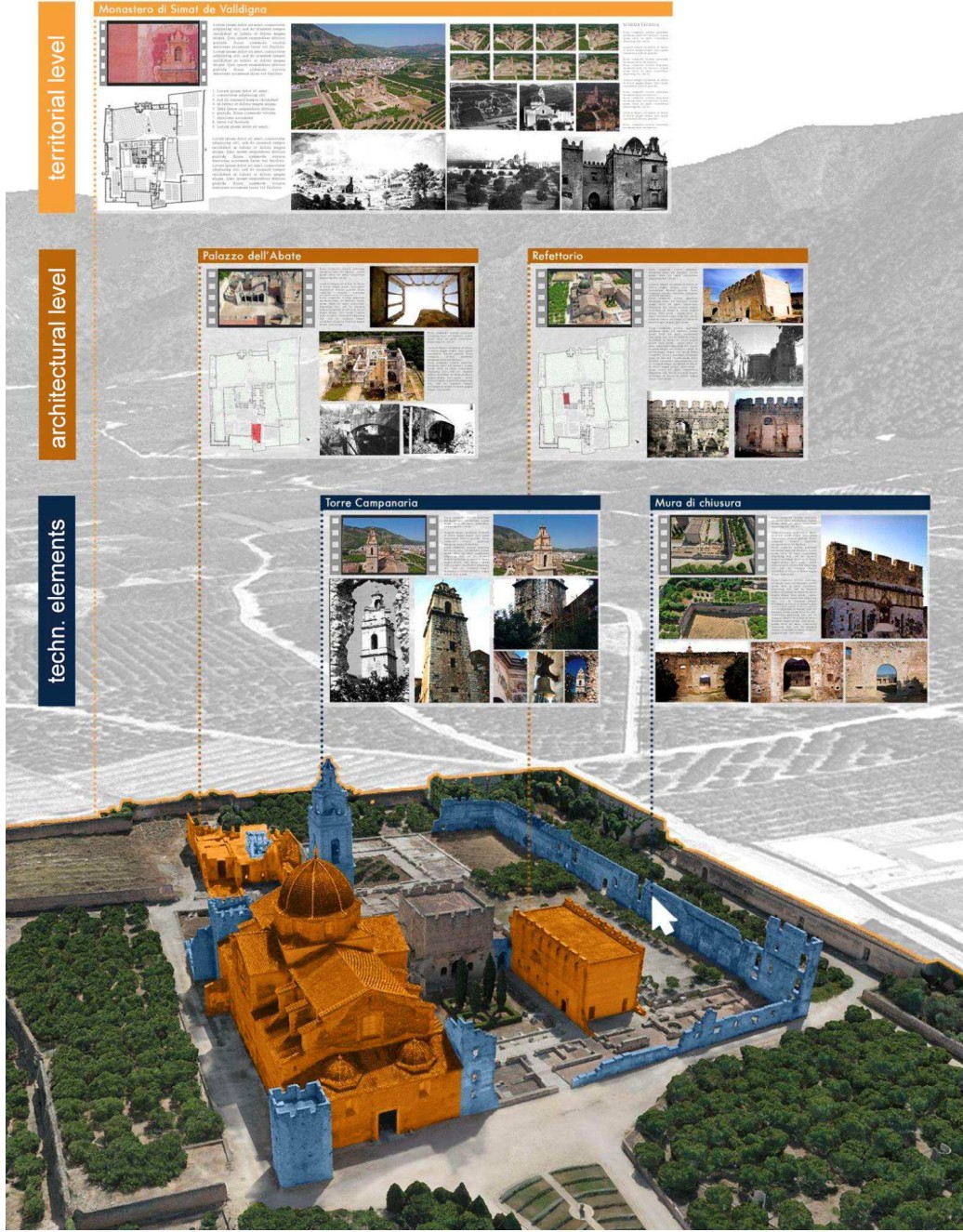

**Figure 10.** Data segmentation for informative models of Simat religious complex on the Jaime I route (data elaboration by G. Porcheddu).

Out of the 50 surveyed and recorded sites, 3D models have been developed for 12 case studies, aimed at creating a comprehensive database of the architectural, constructional, and geometrical components of the current state of various monuments. Additionally, a phase currently underway within the project involves modeling historical reconstructions of some of the previous configurations of the fortified structures. This investigation is being conducted by analyzing past drawings and photographs present in historical archives, as well as through an examination of the current object's configuration. This includes analyzing data produced by 3D surveys and studying existing marks on historical walls and planimetric developments.

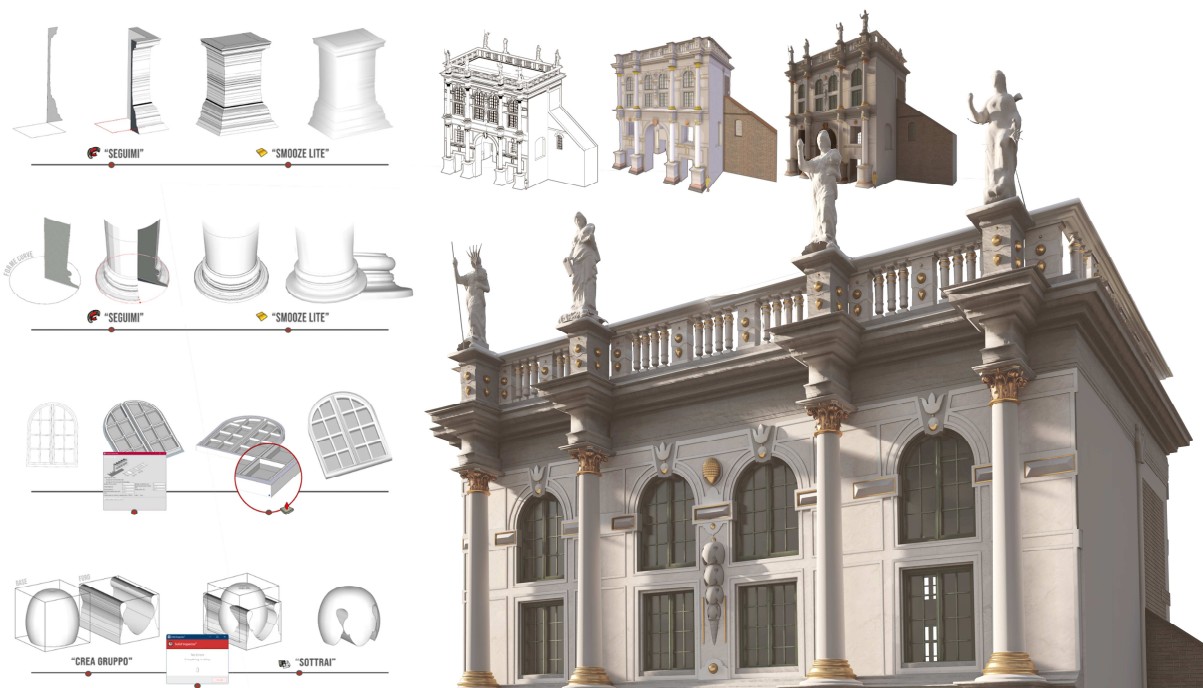

**Figure 11.** Three-dimensional modeling in Rhinoceros, realized from 2D drawings, for the Golden Gate (Gdańsk) (data elaboration by F. Picchio, A. Dell'Amico, S. Bonanomi, C. Caroli, G. Venanti).

In particular, the team is focusing on reconstructing the image of the city's access system, represented by monumental gates, as well as the defensive walls (boundary walls and Renaissance bastions), the transformation of which has significantly influenced the overall cityscape. Once again, the NURBS modeling phase interacted with the BIM and GIS approaches, aiming to develop models of various historical phases that can converge within the interactive platform of the route. This aims to provide an understanding of the transformations undergone over the centuries.

### 3.2. Interactive Informative Platform

With the aim of complying with the open accessibility features of the results required by Europe regarding H2020 projects, each digital twin realized for the monuments of the three cultural routes will be inserted into an interactive and searchable platform to make them visible and usable. The georeferenced 3D database platform provides the main basis for data integration and analysis, allowing comparisons between real objects and their digital representations. Specifically, this involves the realization of spatial information systems (SISs) for cultural heritage applications of GIS technology [39]. This product seems to have several key features that contribute to a range of benefits:

1. Integration of different data: The product connects different types of information content, such as cards, images, parametric models, technical drawings, and analyses. This integration likely allows for a more comprehensive understanding of the architectural sites and their spatial context.
2. Significance of the spatial component: By combining spatial data with other architectural information, the product can help researchers and users better comprehend the importance of the spatial component in their analyses. This could lead to insights into the relationships between each architectural element and its surrounding.
3. Accessibility for different users: The product will be designed to be user-friendly and accessible to various types of users, not just experts in the field. This inclusivity could lead to a broader range of users benefiting from the technology and contributing to architectural heritage research.

4.  SIS technology: Integrating SIS technology into the product can enhance the accuracy and efficiency of spatial analyses, that take into account each building with its context.

5.  Increase in knowledge: The product's capabilities likely lead to a deeper understanding of the value and challenges associated with cultural heritage monuments. This increased knowledge could be instrumental in making informed decisions regarding land planning strategies and preservation efforts.

Such a georeferenced 3D database platform seems to offer a powerful tool to explore, analyze, and preserve cultural heritage sites holistically (Figure 12). Its integration of different types of data and spatial information can lead to valuable insights and foster better appreciation of the significance of these monuments in their respective contexts.

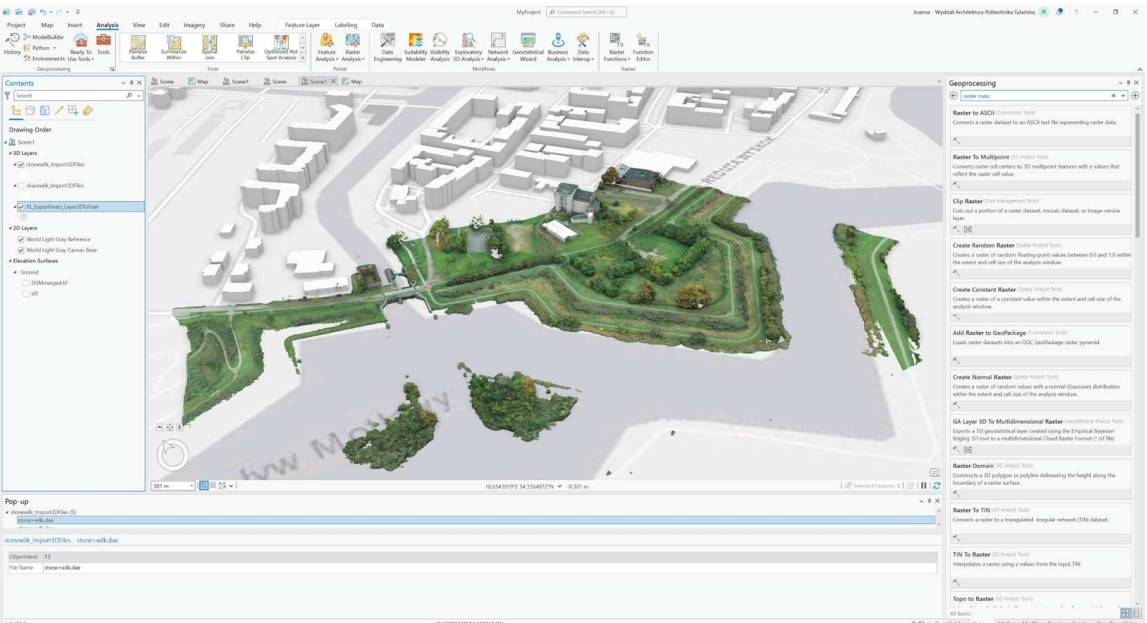

**Figure 12.** Insertion of image-based point cloud of Gdańsk southern bastions into ArcGIS platform (data elaboration by F. Picchio, J. Badach and G. Porcheddu).

This platform is an ambitious project that was initiated during the last year of the project and is still in the development phase. The experimentation conducted thus far has allowed for the design of a multi-level and multi-informational information system, where all currently processed data must find their placement and hierarchical organization in terms of reading and digital system enhancement strategy.

As of now, each of the three platforms (Upper Kama, Valencia, and Gdańsk) has been conceived as an independent and separate web-GIS system.

-   For the Upper Kama project, the three districts that comprise the various analyzed sites will be represented by a platform integrating GIS and BIM systems. The inserted models of monuments and their context (LOD 0 and LOD 1) will be associated with different information (texts related to the toponymy of historical centers and historical and current images), accessible through specific choices in a menu within the developed platform. From here, it will be possible to access a higher level of detail (LOD 2), which will feature interactive elements describing the monumental system of various buildings and their relationships with the context. Clicking on each of these buildings will delve into the detailed BIM of the individual church or monument (LOD 3), from which one can access specific information about the building.
-   For the Valencia route of Jaime I, the platform will not utilize simplified terrain models, but rather a web space built upon a cartographic map of the Province of Valencia. On this map, the locations of various investigated sites will be marked, represented by clickable volumes, in accordance with the geographical layout of the route. The

clickable points within this virtual space will allow interaction with individual reality-based models, optimized in terms of polygon count to ensure ease of management and navigation. Furthermore, these models will be semantically decomposed according to a technological–constructive organization of the historical phases that have impacted the sites. In this manner, by selecting a portion or element of a specific phase, it will be possible to access cards containing textual, graphical, and photographic data that enhance the reading and comprehension of the navigable 3D models.

- For the route of the fortifications in the center of Gdańsk, an idealized platform of the city has been designed, where distances and dimensions of various elements have been rethought in relation to their perception within the urban complex. Houses, streets, and minor urban features have been simplified and placed with variable dimensions based on their perceived significance. Similarly, fortified elements have been modeled based on conducted surveys and proportionally inserted with larger dimensions compared with other elements on the platform. Structured on Unreal Engine 4, the platform aims to emphasize the defensive system of the Polish route. It becomes clickable, enabling access to detailed models of each fortified element. These models have been obtained through mathematical NURBS modeling processes, as well as mesh or parametric modeling for some of the most representative buildings (Figure 13).

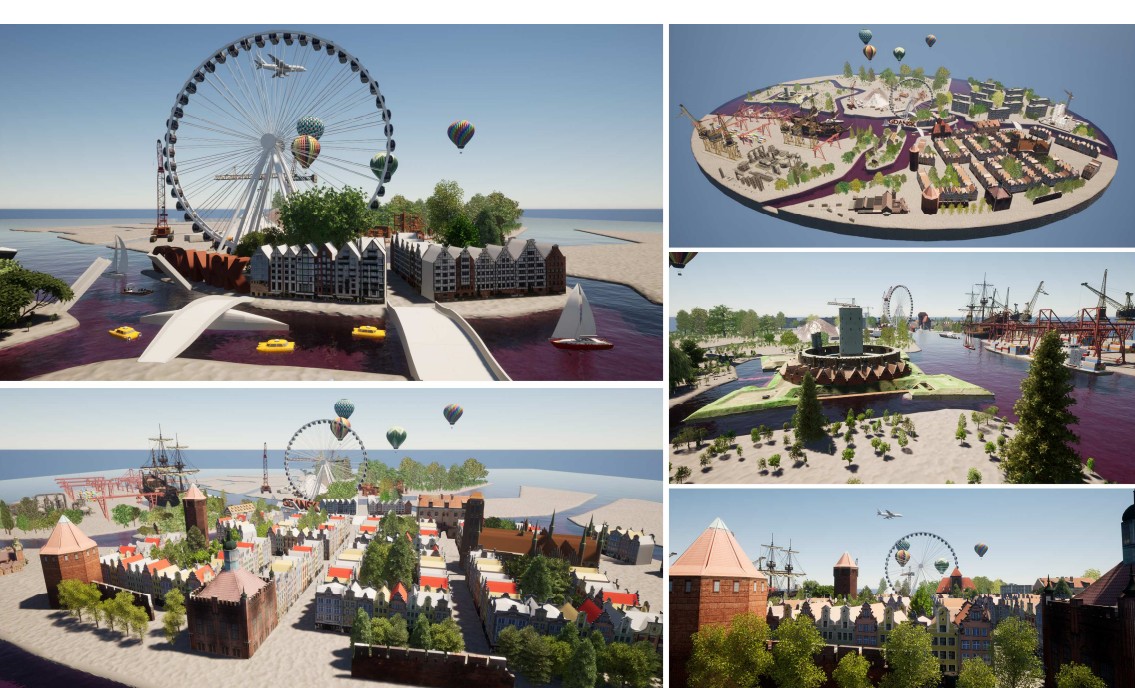

**Figure 13.** An informative platform of an "imaginary" Gdańsk city center (elaboration by S. Parrinello, H. Fu). From this web-GIS platform, it will be possible to access the different 3D models and information elaborated for the cultural route.

The three platforms and all related specific content will be accessible online through the website of the European project (https://www.prometheush2020.eu/, accessed on 26 July 2023). As the system is embedded within the Internet, this must allow accessibility and simplicity for all users, and above all, be independent of any software [40]. A web-GIS platform thus constitutes a new configuration of the route. In particular, on the basis of drone-based reality models, virtual storytelling is realized narrating cultural, architectural, or conservation status aspects of the cultural route, useful reading for non-experts as well as for technicians and experts in the field of conservation and restoration. Each of the route platforms therefore allows different levels of interaction with the monuments on the route: from the more general, where one visualizes and interacts with maps and points; to the

more detailed, where one learns about the history of the artifact by reading its current spatial components and reconstructions of its historical phases.

Another aspect that the project is considering and trying to develop as a further and more in-depth knowledge phase of each site is the possibility of using extended reality, VR, and AR applications [41]. By integrating VR and AR technologies into the project, it becomes possible to extend the reach of the cultural route beyond physical boundaries, enabling people from different parts of the world to explore and learn about the heritage virtually. The use of VR and AR applications in combination with 3D digital models can significantly enhance the development of immersive storytelling experiences along the cultural route, as well as in-depth learning of the object (Figure 14).

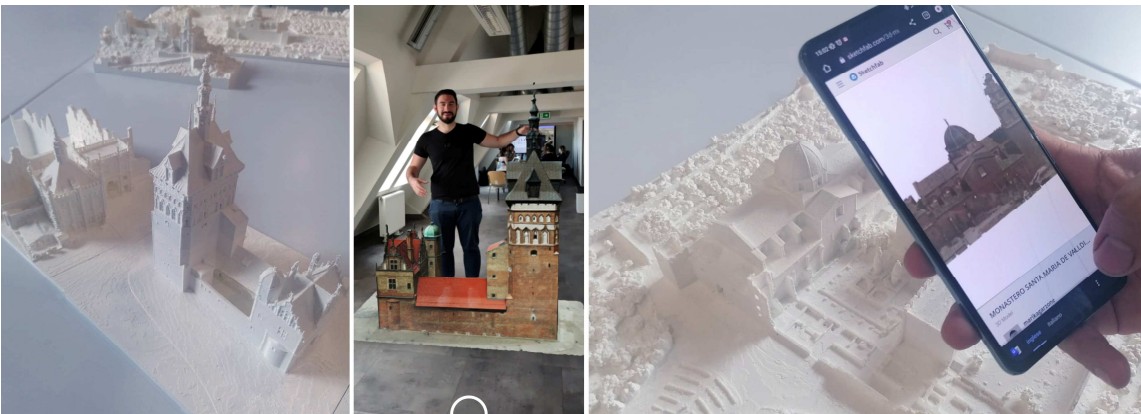

**Figure 14.** A 3D printing for tactile fruition of the Prison Tower monument in Gdańsk. In the middle, AR applications for visualization in 1:5 scale of the monument; on the right, 3D printing and QR codes to access to the textured model on the Sketchfab app.

In particular, some of the 3D models produced (such as the Prison Tower building model from the gate access system in Gdańsk) have been made accessible via mobile devices in augmented reality (AR) mode. This accessibility enables viewing in relation to the real object (the monument) as well as in alternative contexts. The AR visualization application was developed using Unreal Engine 4 software and prepared in apk format for Android devices. On-site access can be initiated through a physical QR code or by displaying the mobile device. This automatically loads and displays the model on the device.

For off-site access, the application "Outdoor guides and explorers" can be utilized. Using the Sketchfab platform, a scale model (ranging from 1:1 to smaller) can be accessed through a link and in a markerless mode. Upon activating the AR function, a short period (around several seconds) is dedicated to recognizing the physical space and establishing a support plane for the virtual placement of the model. Once the model is viewed, users can move around the 3D building model or approach it freely with their smartphones to explore the detailed model from different angles.

Additionally, another mode of access has been developed through a VR application. This mode has been predominantly experimented with for the monumental buildings in the Upper Kama region, given the physical inaccessibility of some monuments and the limited overall understanding of such systems.

In particular, the embedded Sketchfab models were made viewable via oculus. This makes it possible to fully visit the building, both inside and outside, access to which is often forbidden to the public due to structural instability or private use [42]. The future goal is to also include the entire platform designed for the three routes within applications that exploit VR technology, making building–building and building–context relationships of the route visible (Figure 15).

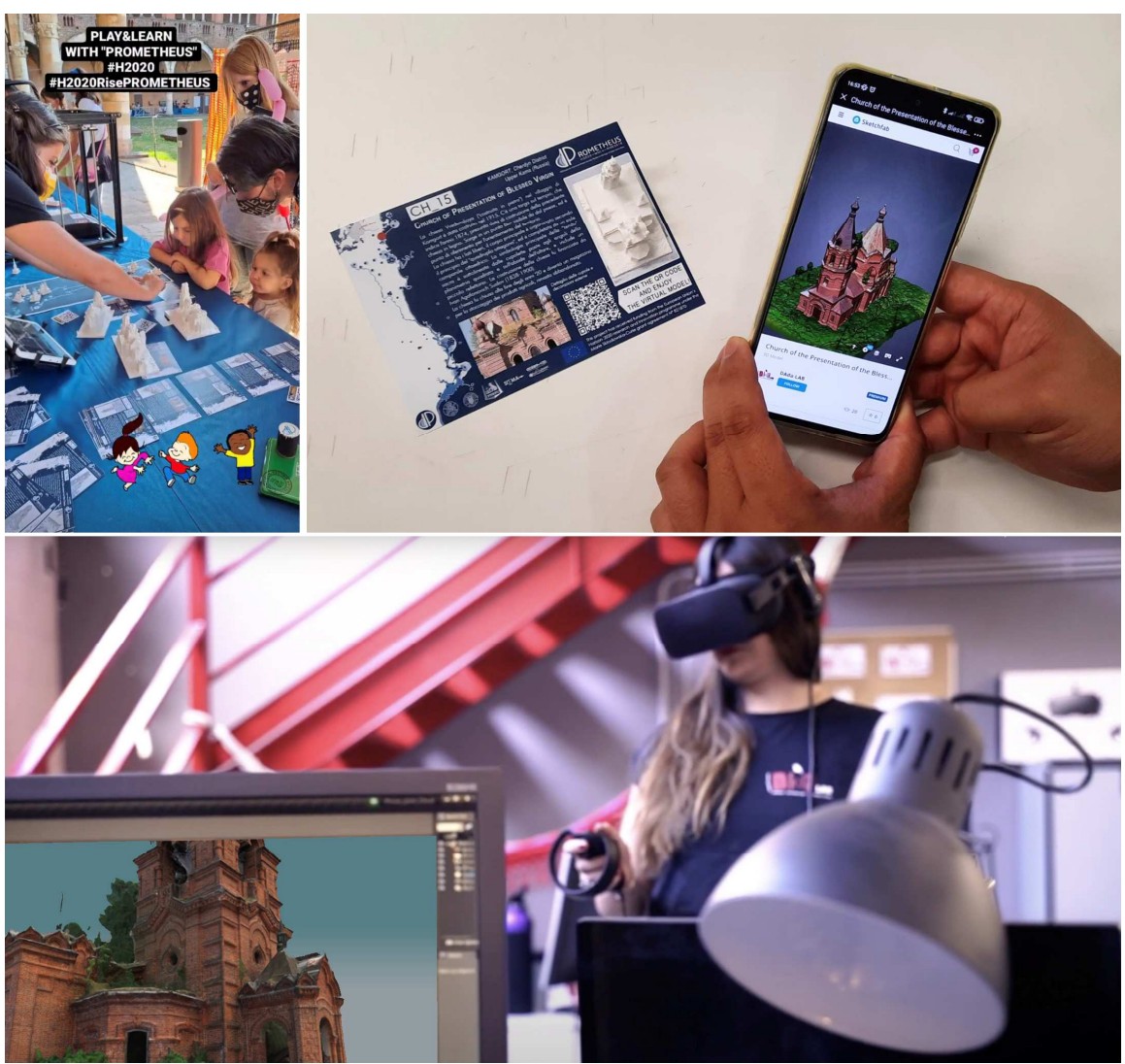

**Figure 15.** Informative cards and 3D models with a QR code linked to Sketchfab, designed for Research Night in Pavia, 2021; below, the design and the test of VR immersive experience with oculus of digital cultural routes of the Upper Kama.

## 4. Discussion

Although the project is in the development phase for these final stages of realization and fruition of the web-GIS platform, it is possible to present some reflections on what has been achieved so far, both from a methodological and a product point of view.

The semantized models serve as visual support for a wealth of available information (historical, geographical, technological) related to each case study of the route. Organizing the elements of the cultural route through segmentation procedures of the 3D database (point cloud or mesh model) becomes a crucial structure for archiving and interpreting the data [43].

This process directly stems from the mental efforts of designing, integrating, and converting a physical element into its digital counterpart, where each digitized element can be part of, or function as, an autonomous, complete, and searchable database. The project's requirement to make research findings accessible through open-access platforms promotes the sharing of these databases, fostering deeper understanding of the different European territories. To ensure easy and immediate access to information, the proposed information system will be developed as Web Apps, allowing the connection of three-

dimensional models (optimized for the web and developed on BIM, NURBS, and mesh modeling software) to related multimedia and iconographic resources [44].

The architectural elements of the cultural routes, differentiated by typology, location, and historical evolution processes, constitute independent databases, each queryable on its own. However, when integrated within the same platform, they can reveal aspects and values that are not always easy to understand [45]. A semantized model, enriched with informative content in this manner, could be configured as a "wunderkammer", a chamber of wonders, containing several configurations of the object itself: historical reconstructions, typological–constructive information, and material and immaterial aspects referring to each part of the semantic model.

With this foundation, the "transition", starting from the survey method and extending to the heritage values, revolves around the interaction between users and the cultural route. Digitizing the itinerary and providing access to "enhanced" content compared with what is available along the route shifts the experience towards the virtual realm, making the itinerary more inclusive [46]. Offering virtual accessibility could attract more informed users, raising awareness about the value of these routes. Consequently, these sites may be considered "protected" areas, recognized as architectural and landscape complexes within specific cultural itineraries.

However, the actions undertaken reveal some critical points in the results obtained and being developed, which also point to some weaknesses in the survey method adopted. The limitations mainly concern the heterogeneity and lack of completeness of some route site information, which makes the platform 'incomplete'. This aspect is mainly attributable to a problem of tight deadlines both in the information acquisition phase and, above all, in the phase of processing this information, which is still not completed. The project, initially dedicated only to the Upper Kama route, was only reformulated in its final year, forcing the adaptability of the method developed in the Russian route to be verified in the other case studies as well. In both the Valencia and Gdańsk cases, the acquisition and processing protocol was readjusted to the different case studies of the routes' sites, inviting a rethinking of more rapidly realizable outputs in order to fit within the project's timeframe. To Upper Kama's parametric information model structure, which required lengthy timeframes and architectural–technological comparisons between similar elements of the same route, two more have been added, envisioning the structuring of information models on much more heterogeneous sites. Defensive monuments and religious sites often presented elements that were not comparable to each other, understandable only when framed with related typological elements of the same geographical contexts, even those not part of the route. As a result, the planning of an investigation protocol designed for a multi-year project but executed over the course of a single year required the simplification and selective acquisition of information, as well as the implementation of pilot tests on some routes rather than the entire cultural itinerary.

On the other hand, we can consider that this forced restructuring of actions and objectives has brought added value to the project. The reimagining of obtaining digital twins more quickly (through rapid survey technologies) and modeling them more efficiently (via reverse-modeling processes) has compelled the team to engage with diverse data and assess multiple solutions, aiming to consistently achieve the custom result that is most functional in terms of timing and available resources, activities perfectly aligned with the project's goal of cultural and experiential exchange.

**Author Contributions:** Conceptualization, S.P.; Methodology, S.P. and F.P.; Investigation, S.P. and F.P.; Data curation, S.P. and F.P.; Writing—original draft, S.P. and F.P.; Writing—review & editing, F.P.; Supervision, S.P.; Project administration, S.P. All authors have read and agreed to the published version of the manuscript.

**Funding:** This project has received funding from the European Union's Horizon 2020 research and innovation program under the Marie Skłodowska-Curie; grant agreement N° 821870.

**Data Availability Statement:** All updated information about the ongoing project is visible on the official website: https://www.prometheush2020.eu/, accessed on 26 July 2023. Part of the output is visible at the following online platform: https://sketchfab.com/DAda-LAB, accessed on 26 July 2023.

**Acknowledgments:** The study is strictly linked to and supported by the research project entitled: Prometheus: "PROtocols for information Models librariEs Tested on HEritage of Upper Kama Sites"—a project for the documentation of the architectural heritage of the monuments within Cultural Heritage Routes. Part of the work has been developed inside International Summer schools, coordinated by University of Pavia, in Russia (2019), Spain (2022), Poland (2023). They have seen professors, researchers, experts, and students from different universities involved participate in the development of digital strategies to build digital twins of cultural heritage.

**Conflicts of Interest:** The authors declare no conflict of interest. The funders had no role in the design of the study; in the collection, analyses, or interpretation of data; in the writing of the manuscript; or in the decision to publish the results.

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
