# Peer review of "Digital Strategies to Enhance Cultural Heritage Routes: From Integrated Survey to Digital Twins of Different European Architectural Scenarios"

_drones, doi:10.3390/drones7090576_

Round 1
Reviewer 1 Report
The paper describes digital documentation and valorization activities. A description of the Prometheus project is provided. Also, the survey methods are introduced (synthetically but with a clear presentation). The use of the data for conservation and touristic enhancements is presented.
The purposes of the paper, of the research presented in it are not clearly exposed. The goals of the investigations performed on specific study cases are provided in the methods section. A common and general goal for the research presented in this paper should be defined and declared in the abstract, and introduction.
The project Prometheus is an exceedingly cumbersome presence that prevents the readers to assess the proposed activities.
ABSTRACT
The purposes of the paper are not clearly exposed in the abstract. Only the "Prometheus" goals are exposed.
INTRODUCTION
The purposes of the paper are not claimed in the introduction. The authors introduce a detailed view of methodologies that allows the investigation, the sharing, and the conservation of cultural heritage. The objectives of this specific paper are not clearly declared.
MATERIAL and METHODS
I think that the description of Prometheus should be moved to the introduction. I suggest authors add a chapter (or a subchapter inside material and methods) titled "sites description" and move in it all the information about the investigated sites
LINES 155 to 174
These are the purposes of the study. There they are referred to each study case, the authors should re-arrange them and move in the abstr. and introd. sections.
CHAPTER 2.2
A few sentences about surveys integration should be added in this chapter
LINE 223-224
remove "...textured Structure-from-Motion (SfM) model to provide..."
CHAPTER 3.1
The results of each investigated case study should be represented in different subchapters, this might help the reader in understanding the innovation and the performed activities.
LINE 241
"...through a geomatic hyper simplification model of the DTM..."
Not clear; what does it means?
LINES 239-261
Which data were used for LOD0 and LOD1? I think the methods described above provide data for LOD2 and LOD3.
If external data are used, the authors should introduced them in the methods chapter.
LINE 355
Is this platform a result of your research (and of this paper)? If not, you should introduce the SIS in the introduction and add a subchapter in the methods.
LINE 358
What's the meaning of [X]?
LINE 396 (VR and AR applications)
The same as above. I think the paper will be more clear to the reader if the result section only contains the specific results of your research.
I highlight in the comments to authors section some minor English revisions
Reviewer 2 Report
The paper falls within a relevant topic in Cultural Heritage digitalisation to support preservation. The PROMETHEUS project is an outstanding project funded by European Commission with promising results. The selected case studies have been exceptionally cured in documentation, representation, visualization and handling. Nevertheless, the paper still require effort to evidence originality in the approaches and methods against current literature. In addition, the overall structure of the text should be revised in order to provide a methodological framework (with methods and equipment) that can be repeatable and scalable for further case studies. Indeed, the introduction explains the importance of “dynamic intervention protocol”, repeated in the Section 2.1 “structuring a methodological protocol”, but the Section 2 does not clearly provide a methodological framework (as a general workflow) that can be furtherly applied – out of the Prometheus project’ context-, and tested in the following paragraphs of the contribution. The re-structuration of the Section 2 (supported by a graphical methodology) could support the reading of the results’ part.
Going in order of appearance, the abstract should be re-structured in order to start with contextual description, research question, methodology, expected results and achieved results, underlining the reasons of originality of the presented work. Authors are suggested to introduce the PROMETHEUS project at the end of the abstract, in order to provide a wider explanation of the answer to previously cited research questions.
I suggest authors to provide acronym extensions the first time they are presented (SLAM, UAVs), since the abstract and through the entire text.
The keywords should be more direct, rationalized and generally conceived.
The Introduction should be wider; readers will expect it to be a framework of the research work (context, research question, research background, gaps-in-knowledge, research statement, expected results).
In addition, the digital twins does not comprehend cultural and historical information only. It is suggested to deep the concept of Digital Twin and explain authors' research work limitations against the complete concept of DT.
The section 2 Material and Methods should structure the methodological framework defined and applied to the three involved case studies. Materials and methods regard the methodological framework, equipment and case studies presentation, before the description of the funding project. The information about the Horizon 2020 project should not be inserted as an ad-hoc paragraph, but it should be introduced at the end of it. Researchers would expect materials and methods here, in order to have a general trace of a methodology that can be repeatable and scalable for other case studies.
“The cultural landscape expressed by the routes is thus narrated through dynamic data-bases that contribute to updating the management and valorisation systems of the built environment [21].” I suggest authors to provide wider explanation of this methodology, involving equipment and limitation scope.
The procedure illustrated in Section 3. Results should be framed in Section 2. Material and methods; for example, the definition of the Level of interpretation and LODs and features, insights, recommendations in methodology should be in Sec. 2, then reported in results (Sec. 3) with measurements and specific description related to the case studies.
“The information components that will qualify the models as Digital twins of specific Upper Kama districts routes are the result of the analysis and instrumental acquisition of the survey activities and the filling of the thematic sheets.” It would be efficient describing the information type to produce the digital twin (both in the research methods as general structure and in results as application).
(Figure 10). It would be useful to see the attached information (previously explained – see comment above) into the model, and how it can be navigated.
As for the Results section, the authors are asked to clarify the structure of the database used for processing the 3D geodatabase and the related web platform. In particular, they are asked to detail in which way the data converge into a spatial information system, explaining the approach used for: i) the integration of multidisciplinary data (images, graphs) into three-dimensional models with different LODs, i.e. the types of information that can be used depending on the LOD of the model; ii) the management of user access in the on-desk and web platform; and iii) the use of data and models at different levels of analysis.
[…] this involves the realisation of what [X] defines à Authors are asked to explain what [X] means
Figure 13 seems not compliant with the content of the sentence; authors are suggested to explain it better.
Figure 15 seems not compliant with the content of the sentence; authors are suggested to explain it better.
I suggest authors to review the Section 4. Discussion. Conclusion paragraph is missing. In addition, it could be useful to provide description of limitation scope and future developments.
TYPOS
End point after citations and figures (i.e NOT BLESARQ, Metaheritage). (Figure 1). ; at the urban scale. [9], and so on through the text).
[…] of signs that renew and characteris the landscape. à characterise
Proofreading is required.

Minor editing of English language required
Round 2
Reviewer 1 Report
Text often stands between figure and caption.
Check fig. 1,2,4,5,6,7,12,13,14,15
